# High-Pressure Supercritical CO_2_ Pretreatment of Apple Orchard Waste for Carbohydrates Production Using Response Surface Methodology and Method Uncertainty Evaluation

**DOI:** 10.3390/molecules27227783

**Published:** 2022-11-11

**Authors:** Lacrimioara Senila, Daniela Alexandra Scurtu, Eniko Kovacs, Erika Andrea Levei, Oana Cadar, Anca Becze, Cerasel Varaticeanu

**Affiliations:** 1Research Institute for Analytical Instrumentation Subsidiary, National Institute for Research and Development for Optoelectronics Bucharest INOE 2000, 67 Donath Street, 400293 Cluj-Napoca, Romania; 2Faculty of Horticulture, University of Agricultural Sciences and Veterinary Medicine, 3-5 Manastur Street, 400372 Cluj-Napoca, Romania

**Keywords:** apple orchard waste, supercritical carbon dioxide pretreatment, RSM, UHPLC, validation

## Abstract

This study’s objective was to separate cellulose, hemicellulose, and lignin after high-pressure supercritical carbon dioxide pretreatment for further valorization. The study investigated the supercritical carbon dioxide pretreatment of apple orchard waste at temperatures of 160–200 °C, for 15–45 min, at a pressure of 10 MPa. Response Surface Methodology (RSM) was used to optimize the supercritical process and to improve its efficiency. The change of functional groups during different pretreatment conditions was examined by Fourier transform infrared (FTIR) spectroscopy. Scanning electron microscopy (SEM) and X-ray diffraction (XRD) confirmed the structural changes in the biomass structure before and after pretreatment. A new ultra-high performance liquid chromatography (UHPLC) coupled with an evaporative light scattering detector (ELSD) method was developed and validated for the determination of carbohydrates in the liquid fraction that resulted after pretreatment. The estimated uncertainty of the method ranged from 16.9 to 20.8%. The pre-treatment of high-pressure supercritical CO_2_ appears to be an effective and promising technique for the recovery of sugars and secondary by-products without the use of toxic solvents.

## 1. Introduction

Agricultural activities generate large quantities of waste worldwide. Apple production and processing generates wastes, including pruning branches, spoiled apples, and pomace [1,2]. Previous research reported the use of lignocellulosic biomass as raw material for the production of ethanol using carbohydrate fermentation, biogas, or fuel for vehicles with fast pyrolysis [3,4]. In recent years, lignocellulosic biomass has been used to separate carbohydrates and convert them into value-added products, such as lactic acid [5]. Lignocellulosic biomass includes wood biomass, residues from forestry operations, agricultural wastes generated by various agricultural activities, and post-harvesting processes [6,7]. Lignocellulosic materials consist of cellulose, hemicellulose, and lignin, which are linked together to form a complex structure. Cellulose is a B-d-glucose homopolymer in which B-1-4-glycoside bonds connect the structural units. Hemicelluloses are polysaccharides consisting of pentoses (xylose and arabinoses), hexoses (mannose, glucose, and galactose), and uronic acids. Hemicelluloses establish hydrogen bonds with cellulosic microfibrils [8]. Due to its branched structure, hemicellulose is relatively easy to hydrolyze. The most critical step in separating important components from lignocellulosic biomass is pretreatment [9].

The separation efficiency of biomass components depends on the pretreatment method used. Supercritical carbon dioxide (CO_2_) is a widely used pretreatment method due to its moderate critical conditions (31.1 °C and 74 bar). The efficiency of supercritical CO_2_ pretreatment depends on temperature, time, and pressure. The mechanism involves the formation of hydrogen bonds that act as a Lewis acid and base. It is recommended to use a co-solvent to increase its polarity. The presence of moisture content and CO_2_ can generate carbonic acid which enhances the hydrolysis of hemicellulose in the liquid phase. Cellulose is soluble in ionic liquid and can precipitate in amorphous form by adding a nonsolvent, such as water or ethanol [10,11,12,13]. Jogi et al. [14] used supercritical ethanol to separate phenolic monomers from birch wood by using the following conditions: 243 °C and 63 bar pressure, 10% moisture content, and iron-based catalysts (5% Fe-SiO_2_).

The response surface methodology (RSM) model can be used to predict the optimal conditions in the variation of different variables. Pinto et al. [15] used RSM for the analysis of the oxypropylation process of lignocellulosic material. Wang et al. [16] reported studies about the use of RSM to optimize the ultrasound-assisted xanthation of cellulose from lignocellulosic biomass for further Pb adsorption. The model used was a Box–Behnken design and it fitted the experimental data and the verification of the variance analysis. Pandey et al. [17] studied the impact of surfactant-assisted acid and alkali pretreatment on lignocellulosic structure and optimization of saccharification parameters by using RSM. Gundapalli et al. [18] investigated the hydrothermal pretreatment of sugarcane bagasse for hemicellulose and lignin separation and used RSM for selecting the optimal operating conditions (water loading and time) in order to increase the digestibility of cellulose.

Several techniques are presented in the literature to analyze carbohydrates separated from lignocellulosic biomass, such as liquid chromatography coupled with a refractive index detector or mass spectrometry, gas chromatography, and UV-VIS spectrophotometry. Generally, the limit of detection (LOD) and limit of quantification (LOQ) is reported, but the uncertainty of the methods is not given [19,20,21]. In this context, the study aimed to develop and evaluate the uncertainty of the ultra-high performance liquid chromatography–evaporative light scattering detector (UHPLC–ELSD) method for the analysis of sugars resulting from the supercritical pretreatment of apple orchard waste (AOW).

The main objectives of the current research were: (i) the application of pretreatment with supercritical fluids assisted by CO_2_ for the separation of cellulosic components; (ii) the improvement of supercritical pretreatment conditions in order to find the optimal conditions through the use of the RSM; (iii) the spectroscopic investigation of raw material, before and after pretreatment by using SEM, XRD, and FTIR techniques; and (iv) uncertainty evaluation of a new analytical method for the analysis of carbohydrates in the hemicellulosic fraction using the UHPLC–ELSD method.

## 2. Results and Discussion

### 2.1. Chemical Composition of Apple Orchard Waste

The chemical compositions of apple orchard waste (AOW) were determined, and the results are presented in Table 1. The AOW contains 58.7% carbohydrates with high cellulose content (32.2 ± 0.07%). The moisture content was 6.96 ± 0.03%, and the total solids were 88.5%. The results showed a high cellulose content (32.2 ± 0.07%) of the AOW.

### 2.2. High-Pressure Supercritical CO_2_ Extraction of AOW

After pretreatment, two phases are formed: the solid phase, which contains cellulose, lignin, and a small amount of hemicellulose; and the liquid phase, which contains monomer sugars, oligomer sugars, and degradation products. The impact of temperature and time of the supercritical process on the AOW conversion of the solid yield and the chemical composition of each phase was studied. The mechanism of AOW supercritical pretreatment involves the following steps: the formation of strong hydrogen bonds with water that migrates on the surface of solid particles, chemosorption on the solid surface, the reaction of hydrogen ions (protons) and xylan at the surface, desorption of xylooligomers or splitting and diffusion of xylooligomers, and transport in the liquid phase.

The composition of the solid fraction after each experiment is presented in Table 2. The solid yield of the solid fraction recovered after the supercritical pretreatment ranges between 72.6–49.3%. It decreases with the increase in temperature and reaction time and it is mainly due to the solubilization of hemicelluloses in the liquid fraction. The composition of the solid fraction was analyzed after each experiment to determine the cellulose, hemicellulose, and lignin content. The cellulose content varies between 37.2–47.2 g/100 g pretreated AOW, and the lignin content varies between 33.0–44.8 g/100 g pretreated AOW. The content of hemicellulose decreases with the increase in temperature and reaction time (from 15 min to 45 min); at 200 °C, hemicelluloses are hydrolyzed almost completely in the liquid fraction. The cellulose and lignin content increase explains its low degradation in supercritical fluids. The amount of cellulose recovered in the solid fraction was high. Lignin has a similar behavior, depending on the temperature and reaction time.

Optimum Condition of the Supercritical CO_2_ Method for Extracting Cellulose Designed by the Response Surface Methodology (RSM).

The RSM technique was used as a statistical tool to evaluate the supercritical pretreatment method [22]. The obtained plots for solid yield, cellulose, hemicellulose, and lignin are presented in Figure 1, Figure 2, Figure 3 and Figure 4. This model evaluates the optimal conditions for the pretreatment to obtain high cellulose content in the solid fraction. The cellulose will be further hydrolyzed and the obtained sugars will be the substrate for bioplastic production.

Based on the contour plots, the theoretical results for solid yield, cellulose, hemicellulose, and lignin content versus temperature and time are presented in Table 3, Table 4, Table 5 and Table 6. The *p*-value was used to observe the significance of the model The degree of freedom is the number of independent variables. The temperature has a low *p*-value of < 0.05 in all experiments, indicating a very high significance. A second-order polynomial was used to measure the correlation between the response and the independent variables.

The analysis of the RSM model for solid yield, cellulose, hemicellulose, and lignin content was verified by measuring the R^2^ coefficient (Table 7). A lack of fit test was carried out. The *p*-values for the lack of fit were <0.05 for solid yield, hemicellulose, and lignin content. The F values were relatively higher than the 95% confidence level, indicating that the model is significant. The higher F suggested that the model is significant for these parameters.

The regression equations (Equations (1)–(4)) are presented below:Solid yield (%) = 509 − 4.48 X_1_ − 0.794 X_2_ + 0.01137 X_1_^2^ + 0.00556 X_2_^2^ + 0.00067 X_1_X_2_(1)
Cellulose = −468.6 + 5.641X_1_ − 0.703 X_2_ − 0.01525 X_1_^2^ + 0.00933 X_2_^2^ + 0.000917 X_1_X_2_(2)
Hemicellulose = 264.5 − 2.51 X_1_ − 0.339 X_2_ + 0.00592 X_1_^2^ − 0.01794 X_2_^2^ + 0.000608 X_1_X_2_(3)
Lignin = −72.3 + 0.896X_1_ + 0.264 X_2_ − 0.001508 X_1_^2^ − 0.00397 X_2_^2^ − 0.001083 X_1_X_2_(4)

The model indicates that temperature and time significantly affect the content of cellulose, hemicellulose, and lignin. The R^2^ value ranged between 93.55% and 95.45%, indicating that the proposed model reasonably adjusted the empirical data. The value of R^2^ must be close to 1 for an ideal model. The model has an R^2^ higher than 93% in all the experiments and the low standard deviation proved the accuracy of the model.

Table 8 presents the optimal solution predicted by the model for each measured parameter. The optimal parameters for obtaining the highest cellulose content are a temperature of 185.8 °C and a reaction time of 45 min. The experimental data obtained after applying the optimum solutions confirm that the experimental and estimated values are close. Becze et al. used a high-pressure extraction process of antioxidant compounds from *Feteasca regala* using RSM, and the values obtained in experiments were confirmed by the RSM model [23]. Wang et al. [16] also used RSM for cellulose content obtained from lignocellulosic biomass by using an ultrasound-assisted pretreatment method. The Box–Behnken design was used.

Based on the results obtained, it can be concluded that RSM is suitable for setting the conditions for improving cellulose production from AOW using high-pressure CO_2_ pretreatment. The model indicated that the temperature and pretreatment time were significant for the content of the compounds that resulted after pretreatment in the solid fraction.

### 2.3. Method Validation and Uncertainty Evaluation for Sugars Analysis by UHPLC-ELCD Detector

The simultaneous analysis of five sugars (xylose, arabinose, mannose, glucose, galactose) from the hemicellulosic fraction separated after the pretreatment of the AOW samples using the UHPLC–ELSD method was developed and validated.

The ELSD performance parameters were optimized to obtain good sensitivity and separation of C5 and C6 sugars. The nebulization temperature, evaporation, and gas flow were tested to obtain the separation of sugars. At a temperature of 70 °C for nebulization and 90 °C for evaporation, the complete separation of sugars was obtained. The unseparated sugars were obtained via nebulization and evaporation at 90 °C. Another factor that affected the separation was the temperature of the chromatographic column. The total time of analysis was 20 min, but in the first eight minutes, the separation of the analyzed sugars occurred. An injection volume of 20 µL provided the best response. Method validation in terms of linearity, detection limit, quantitation limit, accuracy, reproducibility, and recovery was conducted according to the requirements of the EURACHEM Guide [24]. The chromatographic separation of carbohydrates in the sample analysis (conditions T = 200 °C for 45 min) is presented in Figure 5.

Selectivity was demonstrated by evaluating the retention time of blank samples enriched with a known concentration of sugars. The reproducibility was determined by evaluating the retention time variation (%CV). The reproducibility was less than 0.5% in the case of each type of sugar (Table 9). The linearity, LOD, LOQ, and R^2^ obtained for each sugar are presented in Table 9, while the statistical parameters of the calibration curve are given in Table 10. The concentration range for each sugar was 25 (LOQ)−250 µg mL^−1^.

The bottom-up approach was taken into account for calculating the uncertainty. The primary sources of uncertainty contributing to the method are the purity of the standards, the analytical balance used for the preparation of stock solutions, the volumetric flask, pipettes used for the preparation of working solutions, the equipment, the calibration curve, and the repeatability of the method. The purity of the reference material was specified in the quality certificate of the standard. As the quality certificate of the reference material does not specify the uncertainty type, a rectangular distribution of the uncertainty distribution is considered; thus, the uncertainty is divided by a radical of three to transform it into standard uncertainty. The uncertainty of the analytical balance was obtained by composing the uncertainty given by an accredited testing laboratory, and the repeatability standard deviation was used. The following factors were considered for the calculation of the uncertainty of the micro-pipettes: calibration certificate, uncertainty of pipetting at different temperatures, and repeatability. The precision was defined as the standard deviation of the response obtained from the measurement of six parallel samples. S_x0_ gave the uncertainty of the calibration curve calculated as the ratio between the residual standard deviation (S_y_) and the slope (b). The precision was also evaluated as the relative standard deviation (RSD). The expanded uncertainty was calculated considering a cover factor *k* = 2 for a level of confidence of 95% (Table 11). The PG values are accepted because they are lower than the Fisher–Snedecor distribution.

The method developed for the analysis of sugars demonstrated excellent precision and can be used for the analysis of sugars from hemicellulosic fractions obtained after pretreatment.

#### 2.3.1. Chemical Composition of Liquid Samples

The influence of temperature and reaction time on sugar content in a liquid fraction that resulted after pretreatment is presented in Table 12. The results in this study are reported with the expanded uncertainty (*k* = 2) for sugar analysis and standard deviation for furfural and 5-hydroxymethylfurfural (HMF) content.

The composition of the sugars in the filtrate recovered after pretreatment shows that pentoses are much more sensitive to water and CO_2_ treatment under the pretreatment conditions compared with hexose sugars (glucose, mannose, and galactose). The results show a higher arabinose content than other sugar types. The sugars that resulted after the hydrolysis of hemicelluloses are mainly pentoses (xylose and arabinose). After the AOW pretreatment, about 7% of sugars were obtained for the pretreatment performed at 160 °C, 10% at 180 °C, and 8% at 200 °C. For the same reaction time, the sugar concentration increases as the temperature increases. This occurs until a reaction time of 30 min, after which it begins to decrease up to a 45 min reaction time. Alinia et al. [25] obtained the best overall yields for sugar (208 g kg^−1^) with supercritical CO_2_ pretreatment at 185 °C for a 30 min reaction time for wheat straw. The proposed mechanism for the transformation of hexoses into HMF is presented in Figure 6.

The decrease in the 5-HMF concentration could be attributed to the polymerization and hydrolysis of 5-HMF at high temperatures, as well as the production of levulinic acid, humins, and formic acid [26]. The highest furfural content was produced by pretreatment performed at 160 °C for 45 min. The furfural content decreased at high temperatures and time. A possible explanation could be the boiling of furfural at 162 °C. According to Yong et al. [27], supercritical fluid extraction enhances the depolymerization of pentosan. The dissolution of carbon dioxide leads to the formation of carbonic acid which favors acidic conditions for bonding cleavages. Furfural is formed from pentoses contained in the hemicellulose phase. Further, furfural can be extracted from the liquid phase with a solvent (ex. toluene) and used as a chemical. The obtained results suggested that AOW can be a good source for obtaining furfural. Binder et al. [28] obtained furfural from the agricultural residue by converting xylose to furfural with a conversion yield of 53%.

#### 2.3.2. Optimum Condition of the Pretreatment Method for Extracting Sugars in Liquid Fraction Designed by RSM

The final compositions of the sugars were optimized using response surface methodology. The response plot for xylose vs. two different input variables (temperature and time) is presented in Figure 7.

The multiple regression for xylose (Table 13) and the summary reports for the other sugars are presented in Equations (5)–(9).

The regression equations (Equations (5)–(9)) are presented below:Xylose (µg mL^−1^) = −3329 + 36.9 X_1_ + 54.9 X_2_ −0.1011 X_1_^2^ −0.79 X_2_^2^ − 0.022 X_1_X_2_(5)
Arabinose (µg mL^−1^) = −6498 + 72.1 X_1_ + 77.1 X_2_ − 0.1981 X_1_^2^ − 1.124 X_2_^2^ − 0.0314 X_1_X_2_(6)
Mannose (µg mL^−1^) = −702 + 8.56 X_1_ + 17.81 X_2_ − 0.0253 X_1_^2^ − 0.3212 X_2_^2^ − 0.0164 X_1_X_2_(7)
Glucose (µg mL^−1^) = −1110 + 12.60 X_1_ + 5.5 X_2_ − 0.035 X_1_^2^ − 0.0862 X_2_^2^ − 0.00164 X_1_X_2_(8)
Galactose (µg mL^−1^) = −901 + 10.26 X_1_ + 12.18 X_2_ − 0.0286 X_1_^2^ − 0.1990 X_2_^2^ − 0.00417 X_1_X_2_(9)

The *p*-value was used to test the significance of the model for the predicted sugars (Table 14). The lack of fit (in all cases) was smaller than *p* < 0.05, proving that the model was significant.

The values of the coefficient of variation for all parameters show a good fit of the model with the experimental data at a 95% confidence level. The calculated F values (6.85—xylose, 7.64—arabinose, 1.72—mannose, 2.37—glucose, and 2.30—galactose) proved that the model was sufficiently significant. The model can be used only for the fit data between the variables’ minimum and maximum values. The pretreatment temperature was the factor that had the most significant influence on the content of sugars in the hemicellulosic fraction.

### 2.4. Structural Characterization of the AOW before and after Pretreatment

#### 2.4.1. SEM Analysis

To evaluate the structural modification of AOW before and after pretreatment at each temperature, SEM imagining was carried out. In SEM images of raw material, a compressed structure can be observed. During the temperature increase, the fibers are broken and the cellulose structure can be observed. The structure of treated AOW (Figure 8b–d) may be due to the solubilization of hemicellulose in the liquid fraction [29]. The use of water as a co-solvent can create an equilibrium, and some parts of the water’s CO_2_ cannot dissolve it.

The SEM images of the AOW differ depending on the temperature applied to the pretreatment.

#### 2.4.2. FTIR Spectra

The FTIR spectra for the raw and pretreated apple orchard waste is presented in Figure 9.

Fourier transform infrared spectroscopy (FTIR) was used to analyze untreated and pretreated AOW. IR spectroscopy studies provide information on the functional groups present in the structure of raw material and can provide information on changes in the functional groups before and after the pretreatment method. Figure 9b presents the functional groups for the cellulose standard revealing the following band assignments: 3400 cm^−1^ is the OH-stretching vibrations range, 2800 cm^−1^ is CH and CH_2_ stretching, 1500 cm^−1^ is the COH and HCH bending, 1400 cm^−1^ is the CH and COH bending, 1200 cm^−1^ is skeletal vibrations of C-O-C stretching, and 1000 cm^−1^ is the vibration of C-O and ring stretching. The vibration presented at 1739 cm^−1^ was attributed to the valence vibration of the C=O group from hemicellulose. This band almost disappears in the spectrum of treated raw material at different temperatures. The adsorption peaks of lignin were around 1624, 1515, and 1455 cm^−1^. The band at 1371 cm^−1^ was attributed to vibrations of C-H from cellulose. The band at around 1317 cm^−1^ was assigned to C-H vibration from cellulose and hemicellulose. The 1059 cm^−1^ band was caused by CH in plain deformation of C-O from cellulose and hemicellulose. The bands presented at 1427, 1371, 1160, and 1032 cm^−1^ are valence vibrations of cellulose (Figure 9b). There is a visible difference in the FTIR spectrum of the untreated and pretreated AOW, as shown by the absence of adsorption bands or by changing the intensity and profile of certain bands [30].

#### 2.4.3. XRD Analysis

The X-ray diffraction patterns of the raw and pretreated AOW are presented in Figure 10.

Lignocellulosic biomass is mainly composed of polysaccharides (cellulose and hemicellulose) and lignin. Hemicellulose and lignin are amorphous, while cellulose has both amorphous and crystalline components [31]. The XRD technique offers information on the crystalline content. The main crystalline and amorphous peaks arise around 2*θ* = 22° and between 2*θ* values of 15 and 18°, respectively. Two typical diffraction peaks were observed at approximately 15° and 22°, corresponding to the lowest (101) and highest (002) peaks, indicating that type I cellulose was the main form of cellulose in the investigated sample [32]. The peak intensity increases and peaks become narrower with the increase in the pretreatment temperature. The relative fraction of the crystalline component in the cellulose is expressed as the crystallinity index (CrI), calculated according to the Segal empirical equation [33]. The Segal formula is one of the most used techniques for calculating the crystallinity index at the peak height of crystalline materials. The CrI values varied in the following order: 51.4% (raw) < 51.7 (160 °C) < 54.5 (180 °C) < 56.6 (200 °C). The enhanced crystallinity indicates the removal of hemicellulose and lignin [34].

## 3. Materials and Methods

### 3.1. Chemicals and Reagents

All used chemicals were of analytical reagent grade. Acetic acid, dichloromethane, sulfuric acid (98%), sodium hydroxide, methanol, hydrochloric acid, ethanol, toluene, acetone, acetonitrile, and CO_2_ (≥99.8%) were purchased from Merck (Darmstadt, Germany). 5-hydroxymethylfurfural (HMF) and furfural were purchased from Sigma-Aldrich (St. Louis, MO, USA). Standards of D(+)-glucose (99.5%), D(+)-mannose (99.0%), D(+)-galactose (99.0%), D(−)-arabinose (98.0%), D(+)-xylose (990%) were purchased from Merck (Darmstadt, Germany). Sodium chlorite (80%) was purchased from Alfa Aesar GmbH & Co (Karlsruhe, Germany). All solutions were prepared by using ultrapure water (18.2 MΩcm^−1^ at 20 °C) obtained from a Direct-Q3 UV Water Purification System (Millipore, Molsheim, France).

### 3.2. Sample Description

The AOW samples were purchased from the Research Station of the University of Agricultural Sciences “Ion Ionescu de la Brad” from Iași, farm no. 3 “Vasile Adamachi” (Romania). The samples were collected immediately after the cutting operations of apple trees, and they were dried and shredded to a diameter of 0.2 mm.

### 3.3. High-Pressure Supercritical CO_2_ Pretreatment

The AOW was pretreated by supercritical CO_2_ pretreatment using a steel pressure Parr reactor (Parr Instruments, Moline, IL, USA), equipped with a Parr 4523 temperature controller and a 1 L reaction vessel. Approximately 30 g of biomass was placed into the reactor vessel with 300 mL water, and then the mixtures were placed in the autoclave reactor. The reactor contains the following components: an external controller, thermocouple, magnetic stirring system, heating jacket, pressure gauge, gas inlet, pressure valve, and CO_2_ cylinder pressure at 25 °C (6.8 MPa). The CO_2_ amount was introduced in the reactor by using a restrictor valve until the desired pressure (10 MPa), to ensure that only liquid carbon dioxide was present. During the reaction process, stirring was set at 600 rpm to allow better mixing of biomass with CO_2_. The reactor was tightly sealed and heated to a set temperature (160, 180, and 200 °C) and time (15, 30, and 45 min) with a ramp-up time of a few minutes to reach the desired parameters. Pressure increased once the reactor was heated up, and the volume of CO_2_ varied with temperature. After the test was finished, the CO_2_ gas was slowly released using a pressure valve, the heating was stopped, and the reaction vessel was cooled at room temperature. The heating and cooling time varied in each experiment (for the experiment, −200 °C and 45 reaction time); the heating time was 30 min to ensure the equilibrium. The samples were cooled at room temperature, the solid fraction was separated by filtration, and the chemical composition of solid and liquid fractions was determined. The liquid samples were recovered and analyzed for sugar content.

#### RSM Methodology

RSM is used to predict the influence of temperature and time (variables) on cellulose content, hemicellulose content, yield, and solid composition (independent variables) in supercritical CO_2_ pretreatment. The Minitab software was used to optimize the RSM model. The water was chosen as a co-solvent to help the separation. In the supercritical pretreatment of the AOW, the dependent variables were the content of cellulose, hemicellulose, lignin, and solid yields, and the dependent variables were temperature and time. The mathematical model used is detailed in the study conducted by Pinto et al. [15]. The effect of temperature (X_1_) and time (X_2_) on the solid fraction composition that resulted after the supercritical pretreatment was investigated using full factorial. All experiments were conducted in triplicate. Two independent variables were studied at three different levels (Table 15). The central composite design (CCD) was used as a type of response surface modeling. The CCD design was carried out with three levels for individual variables X_1_ and X_2_. The second-order polynomial equation used was presented in Equation (10):
(10)$$Y=\beta_{0}+\sum_{J=1}^{k=2}\beta_{j\;}x_{j}+\sum_{J=1}^{k=2}\beta_{\mathrm{jj}\;}x_{j}^{2}+\sum_{J=1}^{k=2}\beta_{\mathrm{jj}\;}x_{i}x_{j}+Є\;$$
where β_0_ and β_ii_ are the regression coefficients; X_i_, X_ii,_ and X_i_X_j_ are the main, interaction, and quadratic terms, respectively; Y is the responsible variable; k is the number of independent variables.

Optimal conditions were applied in order to verify the model by comparing experimental and predicted values.

### 3.4. Analytical Characterization

#### 3.4.1. Chemical Characterization of Raw and Pretreated Biomass

Ash and moisture were analyzed gravimetrically. The moisture content was determined by drying for 24 h at 105 °C in an oven (Memmert Tip UFE 400 GmbH + Co. KG, Schwabach, Germany) to a constant weight. The ash content was determined after the incineration of samples at 550 °C according to ASTM D1102-84 (2021) [35]. The extractables were determined by extracting the sample with ethanol in a Soxhlet extractor for 6 h. The elemental analysis was determined by a Flash EA 2000 CHNS/O analyzer (Thermo Fisher Scientific, Waltham, MA, USA) [36,37,38]. The content of cellulose, hemicellulose, and lignin was determined according to Teramoto et al. [39]. The holocellulose was determined by the reaction of raw material with NaClO_2_ at 75 °C for 1 h. The cellulose was obtained by the reaction of the previously obtained holocellulose with 17.5% NaOH at 20 °C for 40 min. The hemicellulose content was calculated as the difference between the amount of holocellulose and cellulose. The lignin content was determined as insoluble residue that resulted after the reaction of raw material with 72% H_2_SO_4_.

#### 3.4.2. Chemical Analysis of Sugars Obtained in Liquid Samples

UHPLC (Agilent Technologies, Santa Clara, CA, USA) was used to analyze the liquid fraction that resulted after pretreatment for monosaccharides (glucose, mannose, galactose, xylose, and arabinose), HMF, and furfural. The content of mono-sugars was analyzed by UHPLC (1260 Infinity II), which contains a quaternary pump (Agilent Technologies, G7111B, 1260 Infinity II, Santa Clara, CA, USA), an Agilent Autosampler with an injection valve fitted with a 20 µL sample loop. The separation was performed on a 5 µm Polaris NH_2_ 250 × 4.6 mm (Agilent Technologies, Santa Clara, CA, USA). The column temperature (Agilent Technologies 1290 Infinity II Multicolumn Thermostat, Santa Clara, CA, USA) was kept constant at 30 °C, and the mobile phase flow rate was 0.6 mL.min^−1^. The Evaporative Light Scattering Detector (ELSD) (Agilent Technologies, 1290 Infinity ELSD, Santa Clara, CA, USA) has the following characteristics: nebulization temperature of 70 °C, evaporation temperature of 90 °C, and gas flow of 1.2 SLM. The eluent used was acetonitrile: water (75:25) with a flow rate of 0.6 mL min^−1^ and an injection volume of 20 µL. All the samples were filtered through a 0.45 µM PTFE filter for LC analysis.

The content of furfural and HMF was analyzed according to our recent methods published [40,41].

#### 3.4.3. Strategy for Methods’ Validation

The standards were dissolved in ultrapure water. The working solutions were prepared in eluent (75:25 ACN:water (*v*/*v*)) and used for the UHPLC analysis. The calibration curve was prepared with six appropriate concentrations for each analyte (25, 50, 100, 150, 200, and 250 µg mL^−1^). The stock solutions were diluted with eluent (75:25 ACN:water (*v*/*v*)) and filtered through a 0.45 µm membrane. The limit of detection (LOD) was calculated as 3 × standard deviations of the response given by the analysis of six blank replicates, while the limit of quantification (LOQ) was calculated as 3 × LOD. The precision in the samples was determined at four different spiked concentrations of sugars (25, 100, 200, and 250 µg mL^−1^) and evaluated as relative standard deviations. The homogeneity of the dispersion was evaluated as PG and calculated as a ratio between the standard deviation of the lowest and the highest limits.

The uncertainty measurement was carried out according to the EURACHEM guide [24]. The uncertainty was reported as expanded uncertainty; calculated as *U*_rel_ = *k* × *U*_c_, where *k* = 2 is the coverage factor (for a level of confidence of 95%); and *U*_c_ was the combined standard uncertainty.

### 3.5. Structural Characterization

#### 3.5.1. Scanning Electron Microscopy (SEM)

The AOW samples, before and after pretreatment, were examined using a scanning electron microscope (SEM VEGAS 3 SBU, Tescan, Brno-Kohoutovice, Czech Republic) with a Quantax EDS XFlash (Bruker, Karlsruhe, Germany) detector. The samples were deposited on a double-sided conductive carbon tape on two aluminum stubs and analyzed.

#### 3.5.2. X-ray Diffraction (XRD)

The X-ray diffraction (XRD) patterns were performed using a D8 Advance diffractometer (Bruker, Karlsruhe, Germany), operating at 40 kV and 35 mA, with CuKα radiation (λ = 1.5406 Å), at room temperature. The crystallinity index (CrI) was calculated using the Segal empirical equation (Equation (11)) via a maximum intensity of the 200 lattice diffraction peak (I_002_, 2θ ≈ 22.5°) and the minimum intensity of the diffraction between the 002 and 110 peaks (I_am_, 2θ ≈ 18°) that correspond to amorphous cellulose.
CrI (%) = (I_002_ − I_am_)/I_002_ × 100 (11)

#### 3.5.3. FTIR Analysis

The FTIR spectra of raw and pretreated AOW were recorded using a Spectrum BX II (Perkin Elmer, Waltham, MA, USA) spectrometer in the range of 4000–400 cm^−1^ on 1% KBr pellets with a spectral resolution of 2 cm^−1^, in order to identify different functional groups and monitor changes occurring at the functional groups level during different conversion processes.

## 4. Conclusions

This study investigated the influence of high-pressure supercritical CO_2_ pretreatment on apple orchard waste for separating sugars into cellulosic and hemicellulosic fractions. The contents of solid and liquid fractions that resulted after pretreatment were evaluated. The results obtained in the experiments were compared with the expected results obtained using the RSM model. A new method of sugar analysis has been developed by using the UHPLC-ELSD method. This method is environmentally friendly due to the low consumption of organic solvents and the absence of the extraction step before injection. The uncertainty of the method was calculated for each type of sugar based on a bottom-up approach. The obtained sugars can be used as raw material to produce bioplastics by using specific strains.

## Figures and Tables

**Figure 1 molecules-27-07783-f001:**
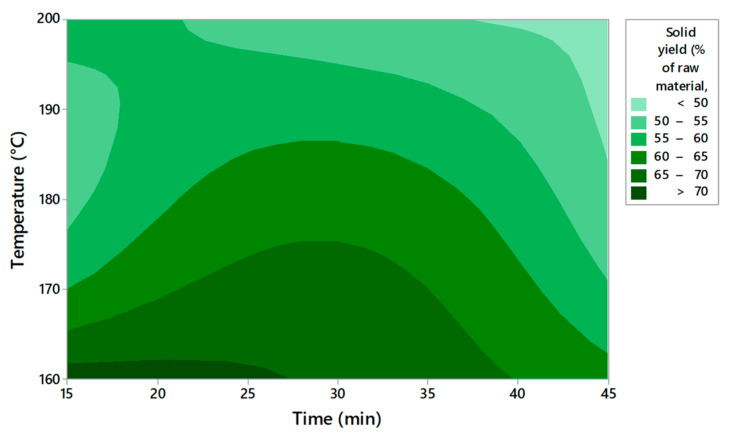
Contour plots of solid yield depending on time and temperature.

**Figure 2 molecules-27-07783-f002:**
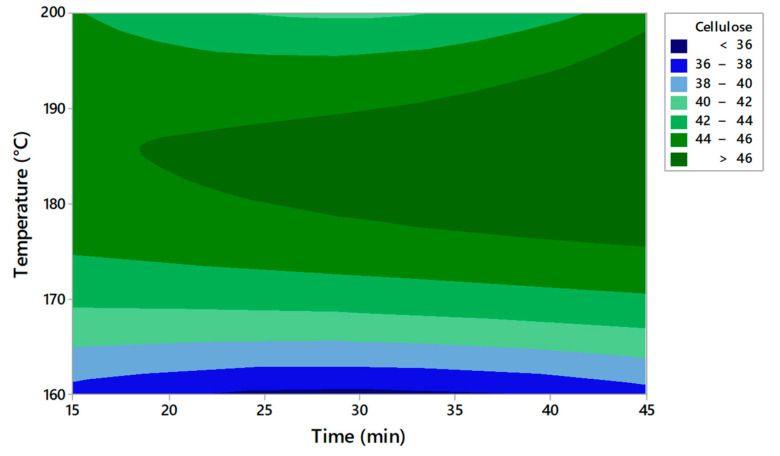
Contour plots of cellulose content depending on time and temperature.

**Figure 3 molecules-27-07783-f003:**
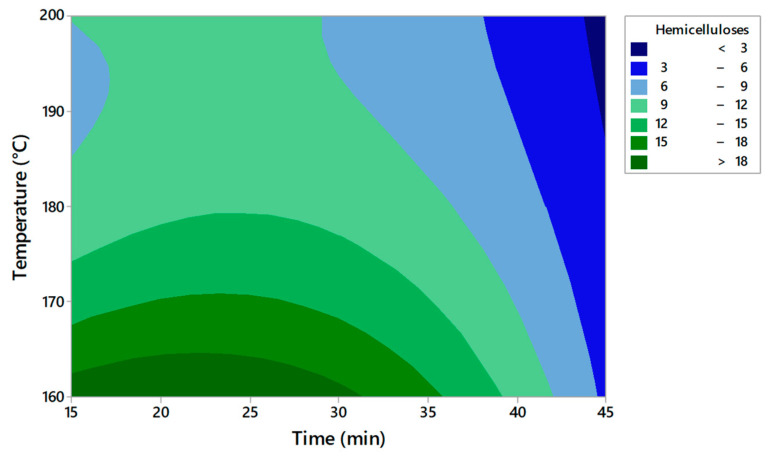
Contour plots of hemicellulose content depending on time and temperature.

**Figure 4 molecules-27-07783-f004:**
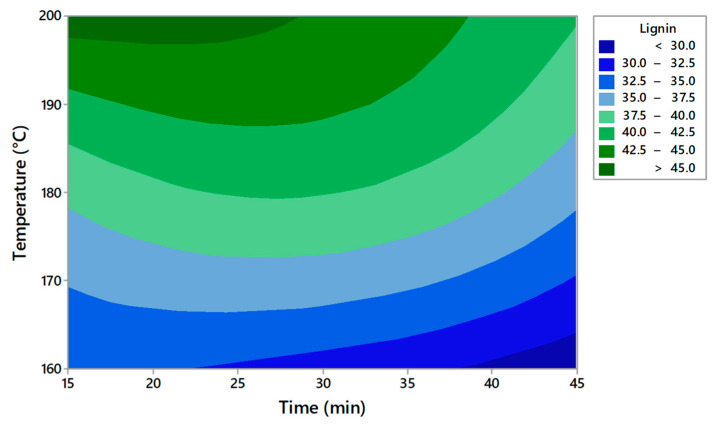
Contour plots of lignin content depending on time and temperature.

**Figure 5 molecules-27-07783-f005:**
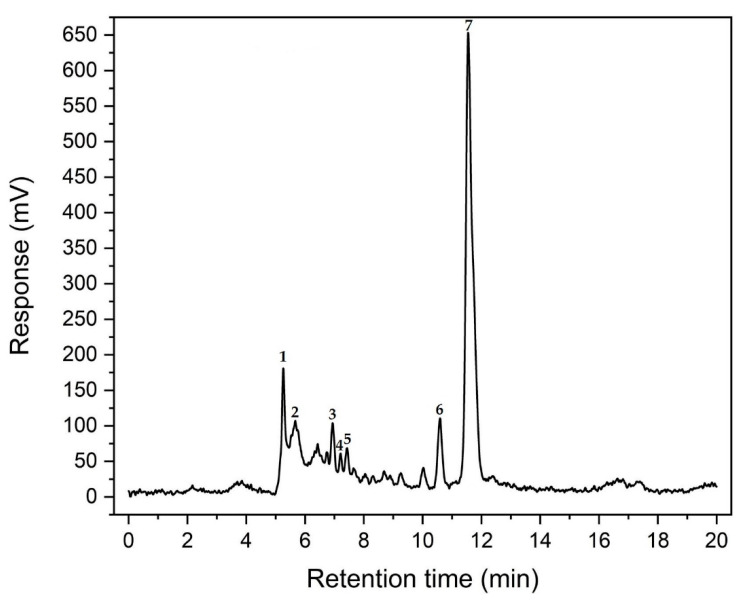
UHPLC–ELSD chromatogram of the carbohydrate sample (1—xylose, 2—arabinose, 3—mannose, 4—glucose, 5—galactose, 6—HMF, 7—furfural).

**Figure 6 molecules-27-07783-f006:**
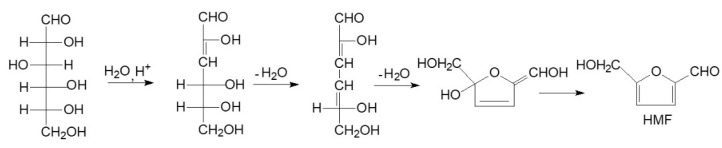
The proposed mechanism for transformation of hexoses into HMF.

**Figure 7 molecules-27-07783-f007:**
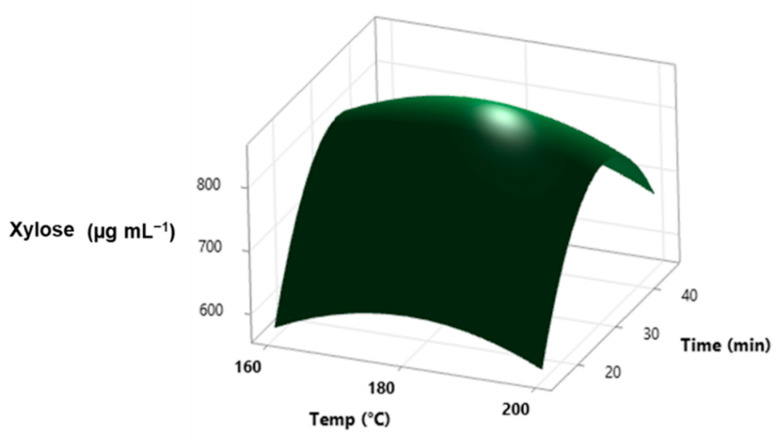
Surface plot for xylose vs. temperature and time.

**Figure 8 molecules-27-07783-f008:**
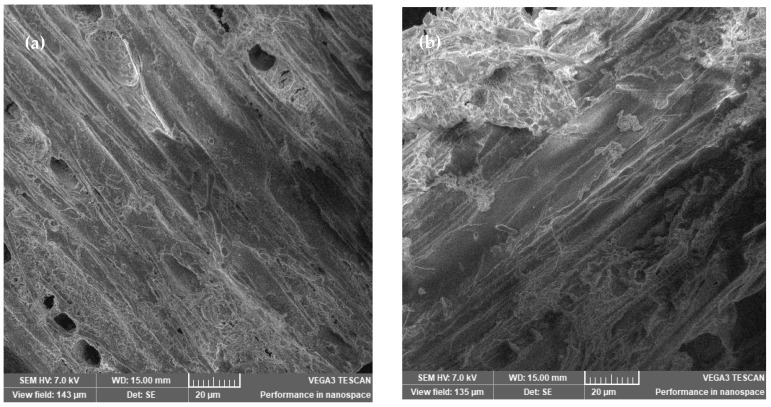
SEM images of (**a**) raw material, (**b**) raw material pretreated at 160 °C for 30 min, (**c**) raw material pretreated at 180 °C for 30 min, and (**d**) raw material pretreated at 200 °C for 30 min.

**Figure 9 molecules-27-07783-f009:**
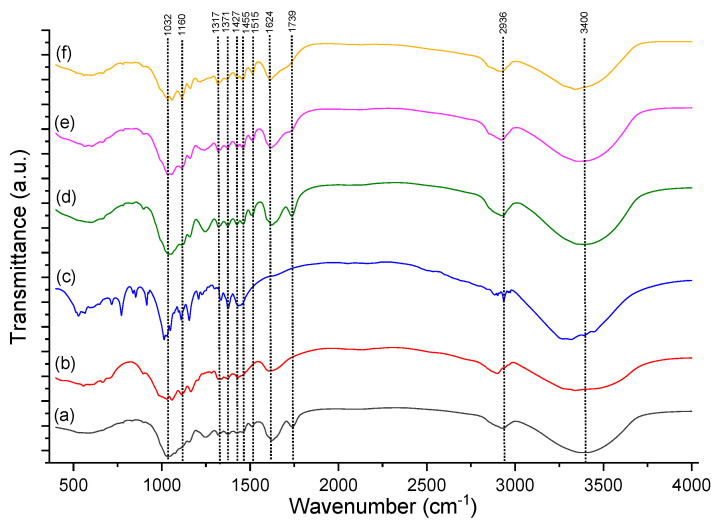
FTIR images of (**a**) raw material, (**b**) cellulose standard, (**c**) glucose standard, (**d**) pretreated biomass at 160 °C for 30 min, (**e**) pretreated biomass at 180 °C for 30 min, and (**f**) pretreated biomass at 200 °C for 30 min.

**Figure 10 molecules-27-07783-f010:**
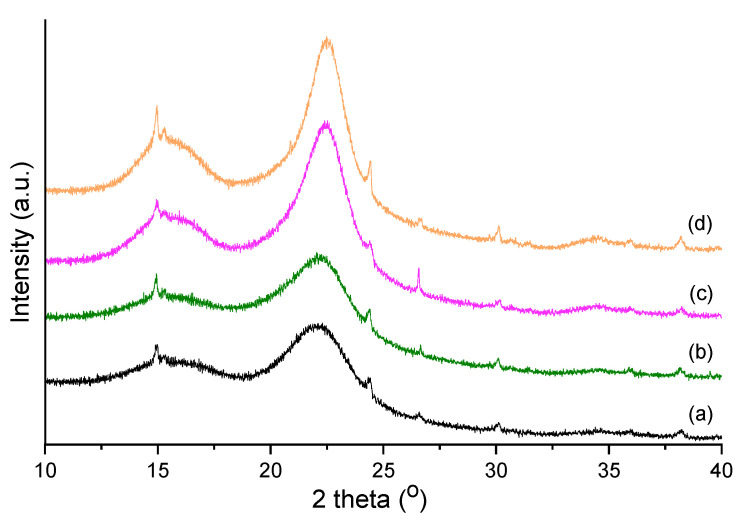
X-ray diffraction patterns of (**a**) raw material, (**b**) pretreated biomass at 160 °C for 30 min, (**c**) pretreated biomass at 180 °C for 30 min, and (**d**) pretreated biomass at 200 °C for 30 min.

**Table 1 molecules-27-07783-t001:** Chemical composition (% *w*/*w*) of the apple orchard waste (AOW).

AOW Component	Amount (%/*w*/*w*)	AOW Component	Amount (%/*w*/*w*)
Lignocellulosic components		Elemental analysis	
Cellulose	32.2 ± 0.07	N	0.78 ± 0.01
Hemicelluloses	26.5 ± 0.05	C	46.7 ± 1.50
Insoluble lignin	24.7 ± 0.07	H	5.79 ± 0.2
Soluble lignin	0.52 ± 0.01	S	0.19 ± 0.01
Moisture	6.96 ± 0.03	Ash	2.04 ± 0.02
Extractables	2.5 ± 0.03		

**Table 2 molecules-27-07783-t002:** Chemical compositions of the apple orchard waste after pretreatment at various temperatures. (Data represents mean ± standard deviation, *n* = 3).

Parameter	Temperature (°C)								
	160 °C– 15 min	180 °C– 15 min	200 °C– 15 min	160 °C– 30 min	180 °C– 30 min	200 °C– 30 min	160 °C– 45 min	180 °C– 45 min	200 °C– 45 min
Solid yield *	72. ± 2.1	63.1 ± 1.8	58.3 ± 2.3	69.2 ± 2.6	53.4 ± 1.5	51.7 ± 2.2	62.2 ± 2.3	51.2 ± 1.8	49.3 ± 1.6
Cellulose **	37.2 ± 1.2	46.3 ± 1.6	44.2 ± 1.5	35.4 ± 1.4	45.3 ± 1.2	41.7 ± 1.3	37.2 ± 1.5	47.2 ± 1.2	45.3 ± 1.3
Hemicelluloses **	19.6 ± 1.8	11.2 ± 0.09	9.1 ± 1.1	18.71 ± 1.2	10.1 ± 0.08	8.8 ± 0.07	5.4 ± 0.06	3.5 ± 0.04	2.2 ± 0.06
Lignin **	33.0 ± 1.2	40.1 ± 1.5	46.2 ± 1.9	31.48 ± 2.1	38.1 ± 2.0	44.8 ± 2.3	28.3 ± 2.4	35.6 ± 2.6	40.2 ± 2.7
Solid compositions	89.9 ± 2.3	97.6 ± 2.1	99.5 ± 2.2	85.6 ± 1.8	93.5 ± 1.7	95.2 ± 1.7	85.6 ± 1.4	93.5 ± 1.7	95.2 ± 1.8

* (% of raw material, dry biomass), ** (% of pretreated biomass, dry biomass).

**Table 3 molecules-27-07783-t003:** Analysis of variance for solid yield.

Source	DF	Adj SS	Adj MS	F-Value	*p*-Value
Model	5	1637.60	327.520	110.02	0.000
Linear	2	1488.17	744.085	249.94	0.000
Temp (°C)	1	993.09	993.094	333.59	0.000
Time (min)	1	495.08	495.076	166.30	0.000
Square	2	148.78	74.389	24.99	0.000
Temp (°C) × Temp (°C)	1	144.39	144.387	48.50	0.000
Time (min) × Time (min)	1	4.39	4.392	1.48	0.238
Two-Way Interaction	1	0.65	0.653	0.22	0.644
Temp (°C) × Time (min)	1	0.65	0.653	0.22	0.644
Error	21	62.52	2.977		
Lack of Fit	3	36.06	12.019	8.18	0.001
Pure Error	18	26.46	1.470		
Total	26	1700.12			

DF—degree of freedom, SS—sums of squares, MS—mean sum of squares.

**Table 4 molecules-27-07783-t004:** Analysis of variance for cellulose content.

Source	DF	Adj SS	Adj MS	F-Value	*p*-Value
Model	5	457.582	91.516	76.21	0.000
Linear	2	212.770	106.385	88.59	0.000
Temp (°C)	1	210.125	210.125	174.98	0.000
Time (min)	1	2.645	2.645	2.20	0.153
Square	2	241.071	120.536	100.37	0.000
Temp (°C) × Temp (°C)	1	216.400	216.400	180.20	0.000
Time (min) × Time (min)	1	24.671	24.671	20.54	0.000
Two-Way Interaction	1	3.741	3.741	3.12	0.092
Temp (°C) × Time (min)	1	3.741	3.741	3.12	0.092
Error	21	25.218	1.201		
Lack of Fit	3	6.092	2.031	1.91	0.164
Pure Error	18	19.127	1.063		
Total	26	482.801			

**Table 5 molecules-27-07783-t005:** Analysis of variance for hemicellulose content.

Source	DF	Adj SS	Adj MS	F-Value	*p*-Value
Model	5	857.407	171.481	122.90	0.000
Linear	2	663.231	331.615	237.67	0.000
Temp (°C)	1	282.903	282.903	202.76	0.000
Time (min)	1	380.328	380.328	272.59	0.000
Square	2	146.336	73.168	52.44	0.000
Temp (°C) × Temp (°C)	1	43.884	43.884	31.45	0.000
Time (min) × Time (min)	1	102.452	102.452	73.43	0.000
Two-Way Interaction	1	47.840	47.840	34.29	0.000
Temp (°C) × Time (min)	1	47.840	47.840	34.29	0.000
Error	21	29.300	1.395		
Lack of Fit	3	22.943	7.648	21.65	0.000
Pure Error	18	6.358	0.353		
Total	26	886.707			

**Table 6 molecules-27-07783-t006:** Analysis of variance for lignin content.

Source	DF	Adj SS	Adj MS	F-Value	*p*-Value
Model	5	749.645	149.929	100.62	0.000
Linear	2	725.549	362.774	243.46	0.000
Temp (°C)	1	621.869	621.869	417.34	0.000
Time (min)	1	103.680	103.680	69.58	0.000
Square	2	15.595	7.797	5.23	0.014
Temp (°C) × Temp (°C)	1	7.114	7.114	4.77	0.040
Time (min) × Time (min)	1	8.481	8.481	5.69	0.027
Two-Way Interaction	1	8.501	8.501	5.70	0.026
Temp (°C) × Time (min)	1	8.501	8.501	5.70	0.026
Error	21	31.292	1.490		
Lack of Fit	3	14.725	4.908	5.33	0.008
Pure Error	18	16.567	0.920		
Total	26	780.936			

**Table 7 molecules-27-07783-t007:** Model summary for RMS model.

	s	R^2^	R^2^ (Adjusted)	R^2^ (Predicted)
Solid yield	1.72	96.32%	95.45%	94.10%
Cellulose	1.09	94.78%	93.55%	91.29%
Hemicellulose	1.18	96.70%	95.91%	94.64%
Lignin	1.22	95.99%	95.04%	93.46%

**Table 8 molecules-27-07783-t008:** Prediction and optimization report obtained from multiple regression.

	Optimal Solutions	Predicted Value (%)	Experimental Value (%)	% of Variation Explained by the Model	*p*
Solid yield	160 °C–15 min	66.42	65.52 ± 1.2	59.75	0.015
Cellulose	185.85 °C–45 min	47.81	48.23 ± 0.8	97.37	0.002
Hemicellulose	160 °C–22.57 min	18.01	16.25 ± 0.3	88.81	0.008
Lignin	200 °C–21.06 min	45.81	43.25 ± 1.1	97.91	<0.001
Solid composition	200 °C–45 min	97.64	96.35 ± 2.6	72.04	0.004

**Table 9 molecules-27-07783-t009:** Linearity, limit of detection (LOD), limit of quantification (LOQ), correlation coefficient (R^2^), and reproducibility of sugars.

Compounds	Retention Time (min)	Reproducibility (*n* = 10) %CV	Regression Equation (*y* = a*x* + b)	R^2^	LOD µg mL^−1^	LOQ µg mL^−1^
Xylose	5.678	0.4	*y* = 18.819*x* − 31.9288	0.993	15.0	25.0
Arabinose	6.133	0.3	*y* = 18.8039*x* − 20.9886	0.992	15.0	25.0
Mannose	7.036	0.3	*y* = 16.9589*x* − 61.1288	0.998	15.0	25.0
Glucose	7.294	0.4	*y* = 39.9770*x* − 192.0075	0.997	15.0	25.0
Galactose	7.676	0.3	*y* = 17.7075*x* − 65.3281	0.998	15.0	25.0

**Table 10 molecules-27-07783-t010:** Statistical parameters of the calibration curve.

Compounds	Sy	S_x0_	Recovery (%)	V_XO_	PG
Xylose	172.23	9.15	99.5 ± 4.2	7.09	2.5
Arabinose	113.80	6.05	107.2 ± 5.6	4.69	3.1
Mannose	81.188	4.79	98.2 ± 4.8	3.71	4.6
Glucose	214.53	5.37	100.1 ± 6.2	4.15	5.5
Galactose	78.770	4.45	104.2 ± 5.5	3.44	5.8

S_y_—residual standard deviation, S_x0_—standard error of the method, V_XO_—coefficient of variation of reproducibility (S_xo_ x 100/X_m_), PG—the limits of working range ($\frac{s_{1}^{2}}{s_{6}^{2}}$).

**Table 11 molecules-27-07783-t011:** Uncertainties of the analyzed sugars.

	Purity (%)	U_c rel_ (mg L^−1^)	U_c_ (mg L^−1^)	U_E_ (mg L^−1^)	U_rel_ (%)
Xylose	99.0	0.104	11.5	22.9	20.9
Arabinose	98.0	0.090	9.9	19.8	18.0
Mannose	99.0	0.085	9.3	18.6	16.9
Glucose	99.5	0.087	9.5	19.1	17.3
Galactose	99.0	0.088	9.7	19.5	17.7

U_c rel_—combined standard uncertainties, U—combined uncertainty, U_E_—expanded uncertainty, U_erel_—relative expanded uncertainty.

**Table 12 molecules-27-07783-t012:** The sugar (µg mL^−1^), furfural (µg mL^−1^), and HMF content (µg mL^−1^) in real samples (mean ± U (*k* = 2), *n* = 5 parallel samples).

Parameter	Temperature (°C)								
	160 °C– 15 min	180 °C– 15 min	200 °C– 15 min	160 °C– 30 min	180 °C– 30 min	200 °C– 30 min	160 °C– 45 min	180 °C– 45 min	200 °C– 45 min
Xylose	581 ± 121	601 ± 125	587 ± 123	784 ± 164	889 ± 185.5	770 ± 161	685 ± 143	708 ± 148	681 ± 142
Arabinose	789 ± 142	815 ± 147	797 ± 144	1064 ± 192	1207 ± 217.5	1045 ± 188	930 ± 168	961 ± 173	925 ± 167
Mannose	238 ± 40.3	246 ± 41.7	241 ± 40.7	320 ± 54.2	363 ± 61.5	314 ± 53.2	281 ± 47.5	290 ± 49.1	279 ± 47.2
Glucose	66.1 ± 11.4	68.4 ± 11.8	66.8 ± 11.6	88.7 ± 15.3	101 ± 17.4	87.1 ± 15.1	77.6 ± 13.4	80.2 ± 13.9	77.2 ± 13.3
Galactose	156 ± 27.6	161 ± 28.5	158 ± 27.9	210 ± 37.1	239 ± 42.2	207 ± 36.6	184 ± 32.5	190 ± 33.6	183 ± 32.3
HMF	22.1 ± 0.07	29.3 ± 0.06	25.3 ± 0.04	25.6 ± 0.05	30.2 ± 0.08	39.5 ± 0.05	32.0 ± 0.7	43.0 ± 0.6	50.0 ± 0.8
Furfural	199 ± 1.5	224 ± 0.8	261 ± 0.9	225 ± 0.21	249 ± 0.34	229 ± 1.1	305 ± 1.3	250 ± 1.8	220 ± 1.2

**Table 13 molecules-27-07783-t013:** Analysis of variance for xylose.

Source	DF	Adj SS	Adj MS	F-Value	*p*-Value
Model	5	242,794	48,559	99.20	0.000
Linear	2	38,885	19,442	39.72	0.000
Temp	1	150	150	0.31	0.585
Time	1	38,735	38,735	79.13	0.000
Square	2	203,375	101,688	207.74	0.000
Temp × Temp	1	9815	9815	20.05	0.000
Time × Time	1	193,561	193,561	395.44	0.000
Two-Way Interaction	1	533	533	1.09	0.308
Temp × Time	1	533	533	1.09	0.308
Error	21	10,279	489		
Lack of Fit	3	5481	1827	6.85	0.003
Pure Error	18	4799	267		
Total	26	253,073			

**Table 14 molecules-27-07783-t014:** Model summary for RSM.

	R^2^	R^2^ (Adjusted)	R^2^ (Predicted)
Xylose	95.94%	94.97%	93.58%
Arabinose	88.89%	86.22%	82.34%
Mannose	87.54%	84.57%	79.29%
Glucose	72.55%	66.02%	57.19%
Galactose	89.13%	86.54%	82.41%

**Table 15 molecules-27-07783-t015:** Independent variables used for response surface studies.

Variables	Symbols	Coded Levels		
		Low Factorial (−1)	Center Point (0)	High Factorial (+1)
Temperature (°C)	X_1_	160	180	200
Time (min)	X_2_	15	30	45

## Data Availability

Not applicable.

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
