# Peer review of "High-Pressure Supercritical CO2 Pretreatment of Apple Orchard Waste for Carbohydrates Production Using Response Surface Methodology and Method Uncertainty Evaluation"

_molecules, 2022, doi:10.3390/molecules27227783_

Round 1

Reviewer 1 Report

Dear Authors,

Thank you for the opportunity of reviewing your manuscript entitled: ” High pressure supercritical CO2 pretreatment of apple orchard waste for carbohydrates production using response surface methodology and method uncertainty evaluation”.

After reading the manuscript I have a couple of comments:

Major concerns:

1.       Chapter 2.2.1  What is the lack of fit for the developed models?

2.       Chapter 2.3.1 – The final composition of the sugars can be also optimized by response surface methodology. 

3.       The developed mathematical model for solid yield indicates that this variable only depends on the process temperature. Hence we have a linear relationship which is in contradiction with Figure 1.

4.       Chapter 3.3  Please elaborate on the description of the experiments. Was CO2 introduced as a liquid or as a supercritical fluid? Whether the reaction system maintain supercritical parameters only concerning CO2? Please give the accuracy of the measuring instruments (temperature and time). What were the heating and cooling time for the system in particular experiments? Could the differences between heating and cooling (experiments in temperatures 200 ° C and 160 ° C) have significantly impacted the final results?

5.       Please review the references carefully; for example, in line 474, the article's subject is not related to separating cellulose from ionic liquids.

Minor concerns:

1.       Different engineering units across the article; for example:   in line 18 (Abstract)  there is a “MPa” as a pressure unit and in line 334 (High pressure supercritical CO2 Pretreatment) “bar”.

2.       Line 97-98 – there are no results on the supercritical process pressure effect in the manuscript

3.       Figures 1, 2, 3 and 4 – different font

4.        Figures 7, 8, 9 – Please add the pretreatment time to the b, c, and d figure label

5.       Line 359 –the sentence is unfinished

Author Response

Dear Reviewer,

Thank you for the constructive comments and suggestions made on our manuscript, ID: molecules-1999282.

The manuscript was revised according to the reviewers’ suggestions. Please find below the answers to the comments and observations. Changes in the manuscript were marked with blue. We hope that the revised manuscript would meet the Reviewer’s expectations.

Dear Authors, Thank you for the opportunity of reviewing your manuscript entitled: ”High pressure supercritical CO2 pretreatment of apple orchard waste for carbohydrates production using response surface methodology and method uncertainty evaluation”. After reading the manuscript I have a couple of comments:

Major concerns:

Q1. Chapter 2.2.1 What is the lack of fit for the developed models?

A1. The Minitab software was used to optimize the model; temperature and time was used as independent variables and solid yield, cellulose, hemicelluloses and lignin as dependent variables. The obtained results were presented in subchapter “2.2.1 Optimum condition of the supercritical CO2 method for extracting cellulose designed by the Response Surface Methodology (RSM)” and the analysis of variance for each parameter analyzed was presented in Tables 3, 4, 5 and 6.  The interaction between temperature and time was included in the model and the regression equations were reanalyzed. The time and temperature were set for each experiment so they would be relevant for the optimization of the model. The lack of fit for each parameter was presented in Tables 3-6.

Q2. Chapter 2.3.1 – The final composition of the sugars can be also optimized by response surface methodology.

A2. The final composition of the sugars can be also optimized by response surface methodology and a new subchapter 2.3.2. “The optimum condition of the pretreatment method for extracting sugars in liquid fraction designed by RSM” was added. The analysis of variace was presented and also the summary of the model.   

Q3. The developed mathematical model for solid yield indicates that this variable only depends on the process temperature. Hence we have a linear relationship which is in contradiction with Figure 1.

A3. Minitab was used to calculate the mathematical model based on the variables (temperature and time) and the results are presented in Tables 3-6. The interaction between temperature and time was presented in the analysis of variance for each parameter.  

Q4. Chapter 3.3 Please elaborate on the description of the experiments. Was CO2 introduced as a liquid or as a supercritical fluid? Whether the reaction system maintain supercritical parameters only concerning CO2? Please give the accuracy of the measuring instruments (temperature and time). What were the heating and cooling time for the system in particular experiments? Could the differences between heating and cooling (experiments in temperatures 200 ° C and 160 ° C) have significantly impacted the final results?

A4. The description of experiments was detailed in 3.3 subchapter “High pressure supercritical CO2 Pretreatment”.  At lines 387-399 the following paragraph was introduced: Approximately 30 g of biomass was placed into the reactor vessel with 300 mL water, then the mixtures were placed in the autoclave reactor. The reactor contains the following components: external controller, thermocouple, magnetic stirring system, heating jacket, pressure gauge, gas inlet, pressure valve and CO2 cylinder pressure at 25°C (6.8 MPa). The CO2 amount was introduced in the reactor by using a restrictor valve until the desired pressure (10 MPa), to ensure that only liquid carbon dioxide was present. During the reacting process, the agitation was set at 300 rpm to allow better mixing of biomass with CO2. The reactor was tightly sealed and heated to a set temperature (160, 180 and 200 °C) and time (15, 30 and 45 min) with a ramp-up time to reach the desired parameters. Pressure increased once the reactor was heated up and the volume of CO2 varied with temperature. After the test was finished, the CO2 gas was slowly released using a pressure valve, the heating was stopped and the reaction vessel was cooled at room temperature. The heating and cooling time varied in each experiment (for experiment 9 -200°C and 45 reaction time), the heating time was 30 min to ensure the equilibrium.”

Q5. Please review the references carefully; for example, in line 474, the article's subject is not related to separating cellulose from ionic liquids.

A5. The reference [11] presented at line 474 “Milavanovic, ST.; Lukic, I.; Stamenic, M.; KamiÅ„ski, P.; Florkowski, G.; TyÅ›kiewicz, K.; Konkol, M. The effect of equipment design and process scale-ul on supercritical CO2 extraction: Case study for Silybum marianum seeds. J. Supercrit. Fluids 2022, 188, 105975.” was replaced with “Seoud, O.A.E.; Kostag, M.; Jedvert, K.; Malek, N.I. Cellulose in ionic liquids and alkaline solutions: advances in the mechanisms of biopolymer dissolution and regeneration. Polymers 2019, 11, 1917.”

Minor concerns:

Q1. Different engineering units across the article; for example: in line 18 (Abstract) there is a “MPa” as a pressure unit and in line 334 (High pressure supercritical CO2 Pretreatment) “bar”.

A1. The unit for pressure was uniformized and MPa was used.

Q2. Line 97-98 – there are no results on the supercritical process pressure effect in the manuscript.

A2. The paragraph presented at lines 99-100The impact of temperature, pressure and time of the supercritical process on the AOW conversion of the solid yield and the chemical composition of each phase was studied.” was modified with “The impact of temperature and time of the supercritical process on the AOW conversion of the solid yield and the chemical composition of each phase was studied.”

Q3. Figures 1, 2, 3 and 4 – different font

A3. The font of Figures 1, 2, 3 and 4 were uniformized (Palatino Linotype 9).

Q4. Figures 7, 8, 9 – Please add the pretreatment time to the b, c, and d figure label 5.

A4. The pretreatment time was added to Figures 8, 9 and 10 (now in the revised paper).

Q5. Line 359 –the sentence is unfinished.

A5. There was a mistake. The sentence was deleted.

Reviewer 2 Report

In the present manuscript, the supercritical carbon dioxide pretreatment of apple orchard waste was employed to separate cellulose, hemicellulose and lignin. There are some comments for this manuscript in this form.

1. Please check the paper grammatically. “243°C” should be “243 °C” in line 59, 6.96 ± 0.03 % should be “6.96 ± 0.03%” in line 91, line 261-262, line 272, line 325, etc.

2. A new column of elemental analysis should be added in Table 1.

3. The title of Figure 1-4 should completely describe the content of the figure, rather than simply listing the conditions, e.g., the combine impact of A and B.

4. In Figure 7, a, b, c, and d should be placed above the SEM picture to optimize typesetting, and pictures should be combined instead of simply listing four pictures.

5. It is suggested to mark the important functional groups and their positions in Figure 8.

Author Response

Dear Reviewer,

Thank you for the constructive comments and suggestions made on our manuscript, ID: molecules-1999282.

The manuscript was revised according to the suggestions. Please find below the answers to the comments and observations. Changes in the manuscript were marked with blue.

In the present manuscript, the supercritical carbon dioxide pretreatment of apple orchard waste was employed to separate cellulose, hemicellulose and lignin. There are some comments for this manuscript in this form.

Q1. Please check the paper grammatically. “243°C” should be “243 °C” in line 59, 6.96 ± 0.03 % should be “6.96 ± 0.03%” in line 91, line 261-262, line 272, line 325, etc.

A1. All the modification were carried out according to the suggestion.

Q2. A new column of elemental analysis should be added in Table 1.

A2. In Table 1, a new column was inserted.

Q3. The title of Figure 1-4 should completely describe the content of the figure, rather than simply listing the conditions, e.g., the combine impact of A and B.

A3. The title of Figures 1-4 was completed with the required information. The analysis of variance for each analyzed parameter was provided.

Q4. In Figure 7, a, b, c, and d should be placed above the SEM picture to optimize typesetting, and pictures should be combined instead of simply listing four pictures.

A4. Figures 8 now (a, b, c and d) - figures numbering was placed above the SEM pictures. 

Q5. It is suggested to mark the important functional groups and their positions in Figure 8.

A4. The important functional groups and their positions were marked in Figure 9 (now).

Reviewer 3 Report

The manuscript is generally well-organized but suffers a major drawback. As displayed in line 354 Page 13, the authors tried to use Equation 1 to optimize the model; temperature and time were recognized as independent variables and their contribution of high orders was also considered. However, the proposed models were not satisfied as most P values were not small enough. I wonder why the interactions between temperature and time were ignored in the model. I would recommend a major revision before it can be considered for publication.

Author Response

Dear Reviewer,

Thank you for the constructive comments and suggestions made on our manuscript, ID: molecules-1999282.

The manuscript was revised according to the suggestions. Please find below the answers to the comments and suggestions. Changes in the manuscript were marked with blue.

Q1. The manuscript is generally well-organized but suffers a major drawback. As displayed in line 354 Page 13, the authors tried to use Equation 1 to optimize the model; temperature and time were recognized as independent variables and their contribution of high orders was also considered. However, the proposed models were not satisfied as most P values were not small enough. I wonder why the interactions between temperature and time were ignored in the model. I would recommend a major revision before it can be considered for publication.

A1: Thank you for the observations. The Minitab was used to optimize the model; temperature and time was used as independent variables and solid yield, cellulose, hemicelluloses and lignin as dependent variables. The obtained results were presented in subchapter “2.2.1 Optimum condition of the supercritical CO2 method for extracting cellulose designed by the Response Surface Methodology (RSM)” and the analysis of variance for each analyzed parameter were presented in Tables 3, 4, 5 and 6. The interaction between temperature and time was included in the model and the regression equations were reanalyzed. The time and temperature were set for each experiment so they would be relevant for the model’s optimization.

Round 2

Reviewer 1 Report

Dear Authors,

Thank you for your comprehensive answers. In my opinion, the manuscript is ready for publication in Molecules.